# Recent Advances in the Application of Natural and Synthetic Polymer-Based Scaffolds in Musculoskeletal Regeneration

**DOI:** 10.3390/polym14214566

**Published:** 2022-10-27

**Authors:** Bing Ye, Bin Wu, Yanlin Su, Tingfang Sun, Xiaodong Guo

**Affiliations:** 1Department of Orthopaedics, Union Hospital, Tongji Medical College, Huazhong University of Science and Technology, Wuhan 430022, China; 2State Key Laboratory of Materials Processing and Die & Mould Technology, School of Materials Science and Engineering, Huazhong University of Science and Technology, Wuhan 430074, China

**Keywords:** polymers, biomaterials, bone, skeletal muscle, tissue engineering, musculoskeletal regeneration

## Abstract

The musculoskeletal system plays a critical role in providing the physical scaffold and movement to the mammalian body. Musculoskeletal disorders severely affect mobility and quality of life and pose a heavy burden to society. This new field of musculoskeletal tissue engineering has great potential as an alternative approach to treating large musculoskeletal defects. Natural and synthetic polymers are widely used in musculoskeletal tissue engineering owing to their good biocompatibility and biodegradability. Even more promising is the use of natural and synthetic polymer composites, as well as the combination of polymers and inorganic materials, to repair musculoskeletal tissue. Therefore, this review summarizes the progress of polymer-based scaffolds for applications of musculoskeletal tissue engineering and briefly discusses the challenges and future perspectives.

## 1. Introduction

The musculoskeletal system consists of muscles, bones, cartilage, tendons, ligaments, and other connective tissues, which play a vital role in providing a physical scaffold and movement to the mammalian body [1]. The number of people with musculoskeletal conditions is rapidly increasing owing to global population growth and aging. Approximately 1.71 billion people worldwide suffer from musculoskeletal disorders, and they are the largest contributors to the demand for rehabilitation services, according to the data provided by Lancet [2]. These diseases severely limit mobility and dexterity and lead to early retirement, lower quality of life, and a reduced ability to participate in social activities. Musculoskeletal disorders are expected to make more people disabled over the next few decades. This is a big problem for the global economy and public health [3].

The capacity of the musculoskeletal tissue to repair itself spontaneously after injury varies widely and is influenced by the degree of inflammation and damage to the surrounding tissue [4]. Muscle recovery is the fastest, and mild strains can fully recover within a few weeks or less without intervention. However, volumetric muscle loss (VML) caused by war and car accident injuries severely affects endogenous repair capacity, resulting in excessive fibrosis and scar formation [5]. Additionally, most long bone fractures heal on their own, whereas large segmental defects caused by congenital malformations, trauma, or tumor resection fail to heal [6]. The tendons and ligaments take longer to recover and heal less effectively. To date, autografts, allografts, and xenografts are regarded as the gold-standard therapies for musculoskeletal tissue treatment. There are some risks that come with these therapies that make them less useful. Their risks include low availability, donor site morbidity, graft rejection, and the spread of disease transmission [7,8].

In recent years, tissue engineering has provided a new direction for the treatment of musculoskeletal disorders. Tissue engineering is a technique that combines the disciplines of biology, medicine, and materials engineering to reconstruct or repair damaged tissues and organs [9,10]. Scaffold materials, seed cells, and growth factors are important research elements in tissue engineering. From this list, scaffolds mainly assume the function of the extracellular matrix (ECM) and are used to support the framework of cells growing into complete tissue. Seed cells are the source of tissue defect repair, and growth factors can guide and coordinate various activities of cells in the tissue [11,12]. These three main elements of tissue engineering can be used independently or in combination. Theoretically, numerous biocomposite scaffolds can be obtained by combination. However, the design and preparation of biomaterials that meet the requirements of musculoskeletal tissue engineering is a major challenge for current research.

Scaffolds for musculoskeletal tissue repair must mimic natural tissue structures and create an extracellular matrix (ECM)-like environment that facilitates cell and tissue survival, which in turn promotes tissue regeneration and functional restoration [13]. Among the various biomaterials used as scaffolds, biodegradable polymers have great potential. They have favorable biocompatibility and biodegradability, are less toxic to human bodies, and are superior to most metals and ceramics [14,15]. Additionally, polymers have excellent plasticity, and they can be easily molded into desired shapes and sizes. Their chemical composition and structure can also be manipulated to meet the specific needs of patients for musculoskeletal repair [16,17]. However, certain natural polymers are mainly suitable for soft tissues and are difficult to match with the stiffness and stability of tissues such as bones and tendons [18]. Therefore, compounding natural polymers with synthetic polymers and compounding organic materials with inorganic materials to form hybrid materials is an excellent strategy to enhance scaffold materials. This can improve scaffold biocompatibility, mechanical properties, and degradation kinetics [19,20].

The development of polymer processing technologies has advanced the functionalization of polymer materials for tissue engineering [21]. Biomaterial processing technologies such as electrospinning, freeze-drying, and additive manufacturing can produce complex polymer matrices that mimic the biophysical and structural properties (e.g., stiffness, roughness, and topography) of the ECM. Their combination with different cell lines can proliferate and differentiate into the desired tissue [22,23]. These emerging processing technologies hold great promise in promoting musculoskeletal tissue repair.

Polymeric materials have a wide range of promising applications in orthopedics, and various types of polymer-based scaffolds have been developed for musculoskeletal tissue repair, but the relevant reviews are not presented comprehensively enough. Therefore, this review first looks at different types of natural and synthetic polymeric biomaterials for tissue engineering applications, then describes the structural and regenerative properties of bone and skeletal muscle, presents the scaffold requirements for musculoskeletal tissue engineering, and provides a systematic review of polymer-based scaffolds suitable for bone and skeletal muscle regeneration. Finally, this review briefly discusses various fabrication techniques used for polymer processing as well as current challenges and future prospects for musculoskeletal tissue engineering. We expect this review to provide a reference for colleagues engaged in research related to polymeric materials for musculoskeletal regeneration and to promote the progress and development of musculoskeletal tissue engineering.

## 2. Natural Polymers

Recently, various advanced polymeric materials have been used for tissue engineering. Natural polymers include collagen, gelatin, hyaluronic acid, alginate, silk fibroin, and chitosan [24]. As a component of the tissue itself, natural polymers have good biocompatibility and biodegradability [25] and can provide innate bioinformatic guidance for cells, improve cell adhesion, and promote the cellular chemotactic response, thus enhancing the biological interaction between scaffolds and tissues [26].

### 2.1. Collagen

Collagen is the most abundant ECM protein in mammals, accounting for 25–35% of the total human protein weight, and it widely exists in the skin, cornea, bone, cartilage, tendons, and ligaments [27]. At present, approximately 28 different types of collagen have been identified and characterized, of which the most common are type I, type II, and type III collagen, which are called fibrous collagen [28]. Collagen is a complex triple-helix structure that appears in a variety of forms in different tissues and interacts with other ECM components to produce higher-order structures with specific functions [29].

For biomedical applications, collagen is extracted from animals, such as pig or cow skin, rat tail, decalcified bovine bone, and rabbit bone [30]. Collagen, as the basic structural component of connective tissue, maintains its structural and biological integrity [31]. Collagen contains the amino acid sequence used for cell adhesion and the glycine-phenylalanine-hydroxyproline-glycine-glutamate -arginine (GFOGER) sequence [32]. Furthermore, collagen has good hydrophilicity, low immunogenicity, high porosity, easy binding to other materials, excellent cytocompatibility, and desirable tissue regeneration potential [33]. Because of these properties, collagen is considered an ideal tissue-engineering material.

### 2.2. Gelatin

Gelatin is a natural protein obtained via acid hydrolysis or alkaline hydrolysis of collagen. Gelatin is certified as a safe material by the FDA and is widely used in pharmaceutical, food, and other industries [34]. It is an attractive biomaterial for tissue engineering and regenerative medicine because of its structural and biological advantages [35]. For example, gelatin is biocompatible, biodegradable, and contains a large amount of arginine-glycine-aspartic acid (RGD) sequences that promote cell adhesion and growth [36]. Furthermore, gelatin molecules contain many chemically modifiable functional groups that can be modulated by various chemical modification methods or loaded with drugs and used as carriers for the delivery of biological materials [37,38]. Additionally, gelatin has gelatinization properties [39] that allow the preparation of a wide range of gelatin-methacryloyl (GelMA) hydrogels for the design of tissue analogs ranging from the vascular system to bone [40].

### 2.3. Chitosan

Chitosan is a major component of the exoskeleton of crustaceans such as shrimp and crabs. It is the deacetylated form of chitin [41]. Chitosan exhibits good biocompatibility and biodegradability. Under the action of enzymes in the body, chitosan degrades into N-acetylglucosamine and glucosamine. These two monosaccharides are non-toxic to humans and can be completely absorbed by the body [42,43]. Chitosan is the only positively charged naturally degradable polymer that has antibacterial activity [44]. By binding to anions in the bacterial cell membrane or entering the bacterial intracellular space, chitosan inhibits cell wall biosynthesis, disrupts bacterial metabolism, and ultimately kills the bacterium [45]. The cationic properties of chitosan allow it to bind to anions that regulate growth factors and cellular activity and promote cell proliferation and tissue repair [46].

### 2.4. Hyaluronic Acid

Hyaluronic acid is an anionic glycosaminoglycan found in synovial fluid and ECM [47]. Owing to the presence of carboxyl groups in the molecule, hyaluronic acid is negatively charged, highly hydrophilic, and viscoelastic [48]. Hyaluronic acid is one of the major intracellular components of connective tissue and plays an important role in cell growth, migration, and differentiation [49]. Hyaluronic acid also has many physiological functions, including the regulation of water in tissues and matrices, and its backbone contains functional groups (carboxylic acids and alcohols) that can be widely used to build scaffolds with structural and space-filling properties [50,51]. As the metabolic and enzymatic processes of hyaluronic acid take place in living organisms, no harmful substances are produced to affect the body. Therefore, it has been used for drug delivery to aid wound repair, cancer treatment, and regenerative medicine [52,53].

### 2.5. Alginate

Alginate is a polysaccharide isolated from natural brown seaweeds such as Laminaria and Lessonia. It is made up of (1–4)-linked β-D-mannuronic acid (M residues) and α-L-guluronic acid residues (G residues), which are covalently bonded [54]. The biocompatibility of M residues is better than that of G residues, whereas the rigidity of G residues is greater than that of M residues. Depending on the origin of the alginate, the number and sequence structure of M and G residues change, which affects the physical and chemical properties of the alginate [55,56]. As an anionic polymer, the alginate can undergo simple reversible gelation through interactions with divalent cations such as Ca^2+^ and Sr^2+^ to form a hydrogel with a reticular structure. This provides a suitable environment for cell proliferation and differentiation [57,58]. The good biocompatibility, hydrophilicity, and ease of formation make alginate a popular biomaterial in tissue engineering.

### 2.6. Silk Fibroin

Silk fibers are mainly produced by arthropods such as silkworms, spiders, and bees. Silk fibroin is a structural protein of silk fibers [59]. It is a naturally occurring amphiphilic block copolymer consisting of a mixture of hydrophobic structural domains and hydrophilic groups that provide flexibility and toughness to silk fibroin [60]. It is biocompatible and has low immunogenicity. The degradation of silk fibroin produces amino acids and peptides that can be absorbed and used by cells [61]. Additionally, silk fibroin has excellent mechanical properties and easy processing characteristics compared to other natural polymers [62]. Silk fibers have been used as sutures in biomedical applications for many years. They are attractive biomaterials that can be used to heal skin wounds, vascular nerve regeneration, and repair tendon, ligament, and bone tissue [63].

## 3. Synthetic Polymers

Compared to natural polymers, synthetic polymers can avoid triggering immune reactions in the human body and can be designed to functionally modify the polymer material to meet the desired function of the biomaterial without changing its intrinsic properties [64]. Polycaprolactone (PCL), poly (lactic acid/L-lactic acid) (PLA/PLLA), poly (glycolic acid) (PGA), poly (lactic-co-glycolic acid) (PLGA), poly (Ethylene Glycol) (PEG), and polyetheretherketone (PEEK) are the most studied synthetic polymers [65,66].

### 3.1. Poly (Caprolactone)

Polycaprolactone (PCL), one of the most common synthetic polymers, is a biodegradable polyester. PCL is a semicrystalline polymer with a melting point of approximately 60 °C and a glass transition temperature of −60 °C, which provides it with good thermoplasticity and molding processability [67,68]. PCL has the advantages of high histocompatibility, high permeability, a relatively slow degradation process, and no toxic by-products. Thus it can be used as a drug delivery carrier [69]. PCL can also be copolymerized with different polymers to obtain the best properties and structures for tissue engineering repair scaffolds such as bone and tendon [70,71].

### 3.2. Poly (Lactic Acid/L-Lactic Acid)

Poly (lactic acid) (PLA) is a hydrophobic, biodegradable polymer composed of lactic acid and is derived from corn starch or other grains [68]. There are three conformations of PLA, poly-L-lactic acid (PLLA), poly-D-lactic acid (PDLA), and poly-D,L-lactic acid (PDLLA) [72]. Compared to PDLA, PLLA has higher crystallinity and chemical stability. The degradation product L-lactic acid is harmless to the human body, while that of D-lactic acid is harmful [73]. Among tissue-engineered degradable polymers, PLA/PLLA has better mechanical properties, biodegradability, and histocompatibility [74,75]. However, PLA tends to degrade and produce acids, which can easily cause inflammatory reactions. Thus, PLA is suitable for fabricating composites to enhance the bioactivity of scaffolds and achieve a good therapeutic effect [76].

### 3.3. Poly (Glycolic Acid)

Poly (glycolic acid) (PGA) is an aliphatic polyester synthesized from glycolic acid by condensation or ring-opening polymerization [77]. PGA has been used as a biodegradable wound suture because of its better fibrillation and excellent bioactivity [78]. Additionally, PGA has higher mechanical strength than other synthetic degradable polymers [79]. However, PGA has a rapid rate of hydrolysis, which makes the material biodegrade very quickly, and it is unsuitable for long-term applications [80]. It must be prepared with other materials to form composite scaffolds for tissue engineering, such as bone and cartilage [81,82].

### 3.4. Poly (Lactic-co-Glycolic Acid)

Poly (lactic-co-glycolic acid) (PLGA) is the most common bioresorbable copolymer, synthesized by the ring-opening copolymerization of two monomers, l-lactic acid (LA) and glycolic acid (GA) [17]. The mechanical and degradation properties of PLGA can be effectively controlled by changing the ratio of LA to GA [83]. Among the synthetic biomaterials, PLGA has been widely used in tissue engineering and regenerative medicine applications. PLGA has good mechanical properties, controlled degradation, and can be tailored to facilitate specific tissue repair [84]. Furthermore, PLGA has excellent biocompatibility and processing properties. Different methods can be used to prepare PLGA-based scaffolds [85]. These scaffolds can be loaded with a variety of bioactive factors to constitute a multifunctional scaffold that can more effectively promote tissue regeneration, such as nerves, blood vessels, skin, cartilage, and bones [86,87].

### 3.5. Poly (Ethylene Glycol)

Poly (Ethylene Glycol) (PEG) is a hydrophilic and uncharged polymer with low protein adsorption and nonimmune properties. The different terminal functional groups of PEG can be copolymerized with the polymer and determine the degradability, mechanical properties, and biological response of the copolymer [88]. PEG is commonly used in the synthesis of hydrogel polymer materials to form highly hydrated polymer gel networks that promote cell growth and adhesion [89]. In recent years, PEG-based hydrogels have been widely used for the tissue regeneration of bone, cartilage, and skeletal muscle [88,90].

### 3.6. Polyetheretherketone

Polyetheretherketone (PEEK) is a kind of aromatic semi-crystalline thermoplastic polymer which contains a repeat unit of one ketone bond and two ether bonds in the main chain. The more ether bonds, the better its toughness [91]. PEEK has properties such as high temperature resistance, chemical resistance, wear resistance, fatigue resistance, high mechanical strength, ease of processing, and light transmission, making it one of the most commonly used materials for orthopedic implants [92]. Additionally, PEEK has the advantages of good biocompatibility and elastic modulus similar to that of normal human bone and can be compounded with glass or carbon fibers to prepare reinforcing materials for bone tissue repair, but the biological inertness of PEEK limits its application in the field of bone repair [93]. Therefore, in recent years, scholars at home and abroad have conducted extensive research on the modification of polyetheretherketone materials, aiming to improve their bioactivity and osseointegration properties in order to play a good medical role in cranial bone repair and vertebral bone repair [94,95,96].

## 4. Applications of Polymeric Materials in Musculoskeletal Tissue Engineering

### 4.1. Bone Regeneration

#### 4.1.1. Structure and Regeneration of Bone Tissue

Bone tissue is a hard connective tissue composed of two main components, collagen and calcium phosphate, which make up the skeletons of humans and other vertebrates [97]. The bone tissue consists of cancellous and cortical bone. The inner cancellous bone structure is spongy and has a porosity of 50–90%. By contrast, the outer layer of the cortical bone is dense in structure and has a porosity of less than 10% [98]. Bone tissue contains three cell types, osteocytes, osteoblasts, and osteoclasts. Osteoblasts mediate bone formation, osteoclasts resorb damaged bone, and osteocytes participate in both bone formation and bone resorption [99,100]. These three types of cells can control the dynamic remodeling, maturation, differentiation, and resorption processes of bone formation. This is achieved through paracrine and endocrine actions, which are important for promoting bone regeneration and maintaining the structural integrity of the tissue [101,102].

#### 4.1.2. Bone Tissue Engineering Scaffold

Based on bone tissue engineering, scaffold materials can improve cell adhesion, proliferation, coordination of interactions between bioactive factors and cells, expression of cell surface receptors, and cell differentiation [103]. As the scaffold material degrades, bone tissue grows into the interior of the graft. Finally, the newly formed bone tissue is used to fill the bone defect and restore bone function, replacing the inanimate material with animate tissue [65,104]. The ideal bone repair material should have the following characteristics: (1) good biocompatibility and controlled degradation rate, and the degradation by-products that can be absorbed or excreted by the body; (2) certain mechanical properties, which can mimic the structural composition and properties of bone tissue, and its mechanical properties should match the tissue at the implantation site; (3) osteoconductivity and osteoinductivity, have a highly interconnected pore network to maintain cell growth and transport of nutrients and metabolic waste, and a material surface that is suitable for bone precursor cell attachment, proliferation, and differentiation, as well as ECM secretion and mineral deposition, and can also induce osteogenic differentiation of MSCs and ectopic osteogenesis [105,106].

#### 4.1.3. Polymeric Materials for Bone Tissue Engineering

Among the polymeric materials used in bone tissue engineering, natural polymers offer excellent performance in terms of biocompatibility. However, their mechanical properties and polymer customization capabilities are limited [107]. Synthetic polymers, on the other hand, do not have the biological cues of ECM, and the by-products of their breakdown can cause acid buildup and inflammation [108]. Since both natural and synthetic polymers have flaws, processing techniques are often used to combine several biomaterials in a certain ratio to make composite materials that meet the needs of scaffold materials for bone tissue engineering scaffold materials [109,110]. The main components of physiological bone tissue are type I collagen and calcium phosphate. The best bone graft to mimic natural bone is a composite of polymers and bioceramics such as hydroxyapatite (HA), β-tricalcium phosphate (β-TCP), and bioactive glass (BG).

##### Polymer-Based Composite Scaffolds with HA

Many studies have focused on the use of HA as a filler to synthesize composites with type I collagen to address the disadvantages of the poor mechanical properties of collagen. Using the freeze-dying technique, Sionkowska et al. prepared a series of porous collagen scaffolds with different HA contents. The addition of HA significantly increased the compression modulus of collagen scaffolds and could be adjusted by changing the cross-linking method, collagen concentration, and amount of HA [111]. Calabrese et al. showed that collagen/hydroxyapatite composite scaffolds greatly boosted the migration, proliferation, and differentiation of human adipose mesenchymal stem cells (hADSC), as well as the production of mineralized ECM [112]. Additionally, the incorporation of HA enables the composite to act as a drug carrier for growth factors such as bone morphogenic protein-2 (BMP-2) and vascular endothelial growth factors (VEGF). Dou et al. used O-carboxymethyl chitosan microspheres (O-CMCS) as a carrier to construct an HA/Col composite microsphere slow-release system with the dual-factor orderly release of BMP-2 and VEGF to promote collagen organization and neovascularization more effectively [113]. Composite materials made of chitosan and HA particles have been widely studied. Zhang et al. used in situ precipitation and freeze-drying methods to prepare chitosan/nHA scaffolds with a three-dimensional oriented structure. Compared with the pure chitosan scaffold, the composite had good compressive strength, a multilayer porous structure, and higher values of osteoblast adhesion ratio and alkaline phosphate activity with increasing HA content [114]. CS/HA scaffolds with hierarchical pore structures were obtained by Li et al. using isothermal plasma treatment and freeze-drying methods. The scaffold had high porosity (≥80%) and swelling rate (≥300%) and exhibited excellent cell viability and biomineralization in vitro [115]. Chitosan is often used in combination with other biopolymers. Shi et al. used dopamine (DA)-modified alginate (Alg) and quaternized chitosan-templated hydroxyapatite (QCHA) to fabricate porous gradient scaffolds. The scaffolds exhibited a high compression modulus (1.7 MPa) and an appropriate degradation rate with excellent osteogenic activity, which could effectively promote bone defect repair [116]. In view of the cationic properties of sodium alginate (SA), SA and CS were made into multilayer slow-release microspheres loaded with VEGF and vancomycin (VAN). Then, it was combined with HA to develop a composite scaffold that kills bacteria and promotes bone tissue regeneration [117]. Silk fibroin (SF) is commonly used in hard tissue engineering to form scaffolds with desirable mechanical properties in complex with HA. Park et al. used nano-hydroxyapatite as a coating for a silk fibroin scaffold with good osteogenic effects both in vitro and in vivo [118]. In a study, McNamara et al. used silk fibroin as a bioadhesive and porogenic agent in the fabrication of HA ceramic scaffolds. It results in silk-fabricated HA scaffolds with high compressive strength and porosity while facilitating the attachment and proliferation of bone marrow mesenchymal stem cells (BMSCs) [119]. Zhang et al. used an electrospinning technique to fabricate a nanofibrous structured HAp/Col/CS composite with high biomimetic and bioactive properties. The scaffold significantly promotes osteoblast proliferation, differentiation, and mineral deposition [120]. Synthetic polymers are often used to fabricate bone tissue engineering scaffolds [121]. Jaiswal et al. used an electrospinning technique to prepare PLLA/gelatin nanofiber scaffolds and obtained PLLA/gelatin/HA scaffolds by compounding with HA. The results showed that the PLLA/gelatin/HA scaffold regenerated the highest bone mineral density (BMD) compared to the PLLA and PLLA/HA groups, and the defect was completely closed at 6 weeks [122]. Gonçalves et al. fabricated a composite scaffold of HA, carbon nanotubes (CNT), and PCL using 3D printing to combine their mechanical properties, electrical conductivity, and bioactivity [123]. Cheng et al. developed a 3D structural scaffold based on PCL using stereolithography and then used a polydopamine (PDA) coating to deposit HA. Compared with PCL, the composite had a higher protein adsorption rate and significantly promoted osteogenic differentiation and angiogenesis in hMSCs [124]. Non-resorbable thermoplastic polymers have excellent physical properties and are showing importance in the field of tissue reconstruction. Abu Baka et al. blended HA into PEEK at 5–40 volume%, respectively, to develop alternative materials suitable for orthopedic applications. The biocompatibility and bioactivity of PEEK-HA composites were significantly improved, but the PEEK-HA tensile properties and fatigue life were affected with the change in HA content. It was shown that PEEK-HA containing 20–30 vol% HA had a tensile strength of 49~59 MPa, a fatigue life of 24.6~32.4 MPa at 10^6^ cycles, and a Young’s modulus of 5~7 GPa, which approximated the mechanical properties of cortical bone [125].

##### Polymer-Based Composite Scaffolds with β-TCP

β-TCP, another representative calcium phosphate bioceramic, exhibits good osteoconductivity and biocompatibility. However, β-TCP degrades faster than HA and can be completely absorbed and replaced by the newly formed bone [126]. Zheng et al. investigated the effects of a composite of β-TCP and collagen (Cerasorb(^®^) orthogonal foam) on femoral condylar defects in rabbits. The results showed that the quantity and quality of newly formed bone induced by the composite were significantly better than those of the blank group [127]. Kato et al. conducted a study to compare the osteoconductivity and biodegradation properties of β-TCP-collagen composites with those of Bio-Oss Collagen^®^ (Osteohealth, Shirley, NY, USA) in a rat cranial defect model. The results showed that the β-TCP composite had better osteoconductivity and biodegradation properties than Bio-Oss Collagen^®^ [128]. Chitosan has been studied as a drug delivery vehicle in tissue engineering for a long time, and its cationic groups have antimicrobial properties. Radwan et al. prepared chitosan-based calcium phosphate composites loaded with moxifloxacin hydrochloride to provide effective mechanical support and site-specific drug release to prevent postoperative osteomyelitis [129]. Gelatin contains free carboxyl groups and is co-blended with chitosan to form an interconnected network via hydrogen bonding. Piaia et al. synthesized four groups of scaffolds, chitosan, chitosan/gelatin, chitosan/β-TCP, and chitosan/gelatin/β-TCP, using different ratios and performed a series of in vitro studies. Among them, the chitosan/gelatin/β-TCP group scaffolds showed up to 70% improvement in mechanical properties compared to the pure chitosan group. Additionally, chitosan/gelatin/β-TCP had better bioactivity and high extracellular calcium deposition, as well as the highest bactericidal effect compared to the other groups [130]. Maji et al. used thermally induced phase separation and lyophilization to prepare porous chitosan/gelatin/β-TCP scaffolds. By incorporating 10 wt% to 30 wt% of β-TCP content, the compressive strength of the composites ranged from 0.8 MPa to 2.45 MPa, and the prepared scaffolds exhibited high porosity (>80%). Additionally, the scaffolds promoted cell proliferation and MSC differentiation in vitro. It also facilitates the formation of new blood vessels and skin tissue regeneration after 2 weeks of implantation into the mouse skin [131]. Park et al. conducted an interesting study in which they evaluated the characteristics and osteogenic induction ability in vitro and in vivo of three groups of scaffolds: SF scaffold, SF/small granule size (300–600μm) β-TCP scaffold, and SF/medium granule size (600–1000 μm) β-TCP scaffold. The SF/small granule size of the β-TCP scaffold showed the best bone regeneration. This is because it has good osteoconductivity and is similar to the components of the actual bone [132]. Novel porous PLGA/TCP/Mg (PTM) scaffolds were fabricated by Lai et al. using a low-temperature rapid prototyping (LT-RP) technique. The PTM scaffolds showed good biomimetic structural design, mechanical properties, and excellent osteogenic and angiogenic properties in the steroid-associated osteonecrosis (SAON) rabbit model [133].

##### Polymer-Based Composite Scaffolds with BG

BG is a silicate-based bioceramic (Na_2_O-CaO-SiO_2_-P_2_O_5_) composed of oxides of silicon, calcium, phosphorus, and sodium. It has excellent bioactivity and biocompatibility compared to CaP-based bioceramics. The Na^+^, Ca^2+^, and PO_4_^3-^ ions released by degradation can promote both osteogenesis and angiogenesis [134]. However, the high brittleness and low mechanical strength of BG limit its application. Therefore, compositing BG with collagen enables scaffolds to have improved mechanical properties and bioactivity [135]. Marelli et al. mixed a plastic compressible dense collagen (DC) gel with nBG to obtain mineralizable hydrogel scaffolds. Carbonated hydroxyapatite (CHA) was formed on the scaffold by day 3 in simulated body fluid (SBF), and the compressive modulus of the scaffold increased 13-fold by day 7. Additionally, Col-nBG effectively promoted MC3T3-E1 metabolic activity, such as alkaline phosphatase (ALP) production, compared to the Col scaffold [136]. Quinlan et al. fabricated a Co^2+^-BG/collagen-glycosaminoglycan scaffold with a high compressive modulus and high porosity (>97%). In addition to the promotion of osteoblast proliferation and differentiation, the Co^2+^-BG/collagen-glycosaminoglycan scaffold has excellent angiogenic capacity [137]. Surface coating is another way to increase the toughness and strength of scaffolds. Keshavarz et al. developed a sodium alginate-coated 58S BG scaffold that significantly improved the mechanical properties and antimicrobial activity of the BG scaffold, as well as the osteogenic differentiation ability of hMSCs on the scaffold [138]. Nawazd et al. used gelatin and Mn-containing mesoporous bioactive glass nanoparticles (Mn-MBGNs) to coat 45S5 BG. The composites showed higher compressive strength, maintained porosity >80%, and enhanced cell proliferation [139]. Dinarvand et al. created PLLA nanofiber structured scaffolds by electrospinning technique and uniformly coated them with HA and BG to obtain implants with efficient bone conduction [140]. Another strategy is to prepare MBG nanofibers (MBGNFs) by electrospinning technique first and then prepare collagen-MBGNF composite scaffolds by the freeze-drying process. This layered biomimetic scaffold has excellent osteogenic and mineralization abilities [141].

The summary of polymer-based bioceramic composite scaffolds for bone regeneration shown in Table 1.

### 4.2. Skeleton Muscle Regeneration

#### 4.2.1. Structure and Regeneration of Skeletal Muscle

The skeletal muscle is composed of muscle fibers, connective tissue, the vascular system, and nerves. The basic structural unit of skeletal muscle is the muscle fiber, which is arranged in parallel and bundled together to form fascicles. Each fascicle is wrapped by perimysium and surrounded by a network of branching capillaries that maintain the metabolic needs of the skeletal muscle. Fascicles and connective tissue form a muscle tissue unit, which constitutes skeletal muscle [146,147]. Muscle-specific resident stem cells, often called satellite cells (SCs), reside between the muscle fiber cell membrane and basal lamina. SCs are responsible for the growth and regeneration of adult skeletal muscle and form the basis for skeletal muscle self-repair. After muscle fiber injury, quiescent SCs are activated and begin to proliferate and differentiate into myoblasts, thereby repairing the damaged muscle fibers [4,148].

#### 4.2.2. Skeletal Muscle Tissue Engineering Scaffold

In the treatment of skeletal muscle injury, skeletal muscle tissue engineering attempts to create a microenvironmental ecological niche that recapitulates key cellular and tissue functions. Thus, the construction of regenerative muscle tissue occurs by regulating cell attachment, survival, and differentiation, repairing and replacing defective or diseased tissue, as well as promoting functional recovery [149]. An ideal scaffold material should meet the following criteria: (1) have desirable histocompatibility without causing immune rejection; (2) have desirable biodegradability to match muscle tissue regeneration rate; (3) have a certain tension and elasticity to match muscle tissue compliance to ensure myotube contractile function; (4) have a certain guiding function to maintain and guide myotube parallel differentiation and growth; (5) have a 3D spatial structure and tissue pores to promote cell adhesion, proliferation, and induce tissue regeneration [150,151].

#### 4.2.3. Polymeric Materials for Skeletal Muscle Tissue Engineering

Among the polymeric materials applied in skeletal muscle tissue engineering, natural polymeric materials have the advantage of good biocompatibility, bioactive signaling cues that enhance cellular behavior, and improved proliferation and differentiation of myogenic cells [152]. They have many functional groups that can bind small molecules and growth factors and deliver them to target tissues. Some synthetic polymers have precisely controlled mechanical properties, chemical compositions, and electrical conductivity. However, poor cell adhesion and unfavorable degradation products have limited their applications [153]. Among the many scaffold materials, hydrogel is a polymer network system with water as the dispersion medium, which is soft and can absorb large amounts of water and maintain a certain shape [154]. Additionally, the mechanical properties of hydrogels, such as elastic modulus, are adjusted to resemble natural skeletal muscle by adjusting temperature and photo-crosslinking. This can promote the development and regeneration of mature skeletal muscle [155]. Therefore, hydrogels are ideal for muscle regeneration. In this section, polymer-based hydrogel materials for skeletal muscle repair and regeneration are discussed.

##### Natural Polymer Hydrogels

Natural hydrogels are ideal for muscle injury repair because they mimic the structural and mechanical properties of natural tissues. They provide a unique microenvironment that facilitates cell survival, proliferation, differentiation, and regenerative repair of skeletal muscle tissue. Most research has been conducted on collagen, gelatin, chitosan, alginate, and hyaluronic acid-based hydrogels for skeletal muscle tissue engineering [156].

Collagen is the main structural component of ECM and has been widely used in muscle tissue engineering. Pollot et al. conducted a study on natural polymer hydrogels to evaluate the mechanical properties and in vitro myogenic ability of type I collagen, fibronectin, alginate, agarose, and chitosan collagen. Based on these results, fibronectin and collagen have the highest myogenic potential and can be used to develop scaffold materials [157]. Li et al. created a new VML model of a single muscle defect in the entire biceps femoris of rats and used vascularized collagen hydrogel constructs inoculated with myogenic cells for VML repair. The constructs exhibited a large number of branched microvascular networks in vitro; however, the in vivo effect was poor due to other factors [158].

Gelatin, a counterpart of collagen, has also been used for skeletal muscle regeneration. Bettadapur et al. cultured mouse skeletal myoblasts using fibronectin (FN) μprinted polydimethylsiloxane (PDMS) structures and micro-formed (μmolded) gelatin hydrogels. The μmolded gelatin hydrogels had a higher myogenic index, myotube width, and myotube length three weeks after differentiation began [159]. Three-dimensional printing can be used to fabricate scaffolds that match the shape of the defect area. Russell et al. used a handheld printer to print gelatin-based hydrogels encapsulated with cells directly into the defect area and cross-linked them in situ. The printing process did not affect the viability of myogenic cells, and after 24 days, the cells differentiated to form multinucleated myotubes, supporting myogenesis and promoting muscle hypertrophy after injury [160]. Seyedmahmoud et al. used gelatin methacrylate (GelMA)–alginate as a bioink to fabricate 3D structural scaffolds. By varying the alginate concentration and crosslinking mechanism, hydrogels that optimally facilitated cell survival, proliferation, and differentiation were obtained. Additionally, the metabolic activity of cells in the bioink was further improved by adding oxygen-releasing calcium peroxide (CPO) particles [161].

Hyaluronic acid hydrogels have good compatibility with skeletal muscle myogenic cells, which facilitates cell adhesion and can be developed as a 3D scaffold material for skeletal muscle repair [162]. Rossi et al. implanted photopolymerized hyaluronic acid hydrogels containing satellite cells (SCs) into mouse tibialis anterior muscle defects and significantly improved the number of newborn muscle fibers and muscle structures, promoting the recovery of muscle function [163]. Goldman et al. used hyaluronic acid hydrogel supplemented with laminin-111 in combination with fragmented muscle graft for regeneration of VML defects in rat tibialis anterior muscle. The regeneration of new muscle fibers increased, and some muscle function was restored [164].

Alginate materials alone are not conducive to cell adhesion due to the lack of cell recognition sites and need to be modified to meet the requirements of muscle tissue engineering applications. Ansari et al. modified alginate hydrogels with RGD to improve their adhesion and then encapsulated gingival mesenchymal stem cells (GMSC) to form 3D RGD-coupled alginate scaffolds. In vitro evaluation showed higher expression of GMSC myogenic differentiation mRNA compared to hBMSC. GMSC also showed greater myogenic regenerative capacity in a subcutaneous implantation model [165]. Additionally, alginate hydrogels can deliver growth factors or drugs and can be used as carriers and support substrates for tissue-engineered cell delivery [166]. Ciriza et al. used an injectable alginate-based hydrogel that transported sodium borate to promote muscle regeneration after injury by stimulating intracellular signaling [167].

Chitosan-derived conductive hydrogels can not only serve as cell delivery carriers but also impart electrical conductivity to hydrogels. This may be a potential direction for improving the muscle regeneration ability of the material. Guo et al. synthesized conductive hydrogels based on dextran-graft-anilinotetramer-4-carbonylbenzoic acid (Dex-AT-FA) and N-carboxyethyl chitosan (CECS) as C2C12 myoblast delivery vehicles. C2C12 cells exhibited a continuous proliferative capacity even after release from the hydrogel, promoting skeletal muscle regeneration in a rat model of volumetric muscle loss injury. An injectable conductive hydrogel is an excellent candidate as a scaffold or cell delivery vehicle for skeletal muscle tissue engineering [168]. Fischetti et al. used chitosan-gelatin hybrid hydrogels as a bioink to fabricate 3D scaffolds by optimizing the printing process and parameters. L929 cells showed good biocompatibility with the hybrid 3D structured hydrogels, and chitosan-gelatin hybrid hydrogels can be developed for the regeneration of various anisotropic tissue, such as skeletal muscle [169].

##### Synthetic Polymer Hydrogels

Due to limitations in biocompatibility and bioactivity, synthetic hydrogels have been less frequently reported in skeletal muscle regeneration studies compared to natural polymer hydrogels [156]. Moreover, synthetic polymer hydrogels alone for skeletal muscle defects cannot achieve the desired repair effect, and they need to be modified to improve their material properties. Therefore, synthetic polymer-based hydrogels can be used as drug or growth factor carriers, and their release may be controlled to help muscles heal [153]. For skeletal muscle tissue engineering, polyethylene glycol (PEG)-based hydrogels are often used as biological factors or drug delivery vehicles. Han et al. designed a series of PEG-based synthetic hydrogels (PEG-4MAL) that delivered muscle satellite cells (MUSC) alone or in combination with MUSC and pro-muscle factor (Wnt7a) for the treatment of tibialis anterior muscle frostbite in mice. The PEG-4MAL hydrogels that were engineered significantly promoted MUSC proliferation and differentiation, as well as muscle fiber hypertrophy [170,171]. Conductive hydrogels have promising applications in skeletal muscle tissue engineering. Xie et al. synthesized an electroactive ductile PLA copolymer (HPLAAT) by copolymerizing a hyperbranched extensible polypropylene cross-ester (HPLA) with an aniline tetramer (AT). Compared to HPLA, HPLAAT significantly promotes the proliferation and myogenic differentiation of C2C12 cells in vitro [172]. Ziemkiewicz et al. first synthesized a PEG-LM111 hydrogel that effectively promoted the proliferation of satellite cells at the injury site. PEG-LM111 hydrogels were mixed with 5% and 10% (*w*/*v*) pure PEG-diacrylate (PEGDA) to fabricate 5% and 10% PEGLM hydrogels, respectively. The 5% PEGLM hydrogels showed superior swelling and porous structure. Moreover, such hydrogels were able to promote myoblast adhesion, survival, and the production of pro-regenerative factors such as VEGF and IL-6 [90].

##### Composite (Hybrid) Hydrogels

To further optimize the various properties of hydrogels, composite hydrogels were prepared by combining natural and synthetic polymers. The ability to use the merits of natural and synthetic materials can advance skeletal muscle regeneration treatment methods [173]. Wang et al. used an electrospinning method to prepare aligned nanofiber yarns (NFY) that mimic the aligned structure of muscle fibers using PCL, silk protein (SF), and polyaniline (PANI). Polyethylene glycol-co-glycerol sebacate (PEGS-M) polymer, a light-curing hydrogel material, was selected to mimic extracellular connective tissue. Finally, a core–shell composite scaffold was prepared using NFY and PEGS-M. The composite scaffold showed good biocompatibility in vitro and was able to effectively induce myogenic differentiation and myotube formation in C2C12 myogenic cells [174]. Carleton et al. conducted a study to compare the regenerative effects of two hydrogels: methacrylic acid-polyethylene glycol (MAA-PEG) and methacrylic acid-collagen (MAA-COL), for the treatment of VML injury in the tibialis anterior muscle of mice. The MAA-COL hydrogel significantly improved the quality and function of regenerated muscle. Moreover, compared with MAA-PEG, the MAA-COL hydrogel not only improved vascularization but also increased innervation [175].

The summary of polymer-based hydrogel materials for skeletal muscle regeneration shown in Table 2.

## 5. Scaffold Fabrication Methods

### 5.1. Solvent Casting and Particulate Leaching (SCPL)

SCPL is one of the most common and simplest methods for manufacturing conventional polymeric scaffolds. By adjusting the porogen/polymer ratio, the scaffold pore size, porosity, and interconnectivity can be controlled [177]. The technique involves first dissolving the polymer in a selected organic solvent, mixing it with a porogen (e.g., sodium chloride, etc.), and casting the mixture into the mold. After evaporation or lyophilization of the organic solvent, the polymer composite is immersed in water to dissolve the porogen and finally, a porous network of scaffolds is obtained [178]. Thadavirul et al. prepared porous polycaprolactone (PCL) scaffold with highly interconnected pores and relatively uniform pore size (378–435 μm) using sodium chloride and PEG as water-soluble porogens [179].

### 5.2. Gas Foaming

Gas foaming is a method of creating porous structures by forming dispersed bubbles throughout the polymer. The base material, the blowing agent, and the adhesive are first mixed and then the mixture is molded into the envisaged shape of the support. The material is immersed in the prepared solution, the blowing agent reacts with the solution to create bubbles inside the polymer structure, and the gas escapes to form a porous structure of the scaffold [98,180]. The pore size is controlled by controlling the mixing ratio of polymer and blowing agent, and the more gas produced, the higher the porosity. Torabinejad et al. fabricated triblock copolymers of PLLA-co-PCL with nHA by gas foaming/salt leaching method using Sn(Oct)_2_ catalyst as initiator and ethylene glycol as co-initiator. The scaffold has excellent porosity and a regular internal pore structure [181].

### 5.3. Thermally Induced Phase Separation (TIPS)

In TIPS, the polymer is first dissolved in a low melting point solvent, then water is added to the solution to produce a polymer-rich phase and a high solvent phase, and subsequently, the temperature of the mixture is adjusted to below the solvent melting temperature, and the polymer-rich phase forms a matrix, while the solvent-rich phase becomes porous due to solvent removal [182]. The properties of the scaffold are controlled by varying the solvent type, polymer concentration and temperature gradient. Ma et al. prepared a PLLA/Hap composite scaffold with a highly porous structure and good osteoconductivity using a thermally induced phase separation technique [183].

### 5.4. Freeze-Drying

Freeze-drying is a method used to dry material/polymer solutions to create porous 3D scaffolds for tissue engineering that can provide an ideal microenvironment for cell culture [184]. The freeze-drying method begins by freezing the solution at low temperatures and then placing the frozen sample in a closed chamber where the pressure is reduced to a few millibars by vacuum. During this drying process, both ice crystals and unfrozen water are removed from the material, resulting in a three-dimensional porous scaffold [185]. Shahbazarab et al. used a freeze-drying process to mix zeatin (ZN), CS, and nHAp in different ratios to obtain porous composite scaffolds with highly interconnected pore-size structures and superior cytocompatibility [186].

### 5.5. Electrospinning

Electrospinning is a popular scaffold fabrication technique that enables the design and fabrication of nanofiber scaffolds similar to natural ECM [187,188]. In this method, the polymer melt or solute is first poured into a syringe with a capillary tube, and under a high-voltage electric field, the polymer fluid overcomes surface tension to flow in the direction of the electric field, resulting in long and thin threads, and finally, nanofibers are stably deposited on the substrate to form the scaffold. By adjusting the distance between the substrate and the syringe, the voltage, and the solution concentration and flow rate, the scaffold pores can be altered to obtain uniform-size fibers [189]. Physical cues in materials can modulate the interaction between cells and materials, with important implications for cell function and tissue regeneration. Choi et al. used an electrospinning technique to mix PCL and collagen to make unidirectionally oriented nanofibers, and the results showed that the oriented nanofibers could guide muscle cell alignment and enhance myotube formation [190].

### 5.6. Additive Manufacturing

Additive manufacturing is commonly used for polymer processing as an emerging biomanufacturing technology. The developed additive manufacturing methods include 3D printing, fused deposition modeling (FDM), and stereolithography (SLA). These methods link polymers with computer-aided design (CAD), computer-aided manufacturing (CAM), laser technology, numerical control technique, and computed tomography (CT) or magnetic resonance imaging (MRI) for scaffold fabrication. Compared to traditional methods (e.g., solvent casting and salt leaching, foaming and phase separation), additive manufacturing allows not only the fabrication of macroscopic shapes that precisely match their interfaces but also the customization of the internal structure of the scaffold (e.g., microscopic morphology, pore size, porosity, pore distribution, etc.) in order to confer its attachment and guidance to cells. In a sense, 3D printing allows the fabrication of different tissue grafts to maintain their functional properties, such as skin, blood vessels, cartilage, and bone. Koo et al. used an extrusion-based bioprinting technique to fabricate collagen/HA constructs loaded with MG63 cells and hADSC. The composites had a meringue-like porous cell-laden structure and exhibited excellent metabolic and osteogenic activities [144].

## 6. Conclusions and Future Prospects

Bone and muscle tissue repair materials made from natural and synthetic polymers have greatly facilitated their application in musculoskeletal tissue engineering. Because of their biodegradability, biodegradable polymer materials can be used to support cell adhesion, proliferation, and differentiation to promote tissue repair. They can also be used to repair tissues while allowing the scaffold to degrade into harmless products and be absorbed by the body. Even though the use of polymeric materials in musculoskeletal repair has shown promising results, several issues remain to be addressed. Vascularization of large implants is challenging for muscle tissue engineering, and achieving good integration with the surrounding tissue of the defect is difficult. In musculoskeletal tissue engineering, bioactive molecules such as growth factors and drugs can regulate cell growth, proliferation, and differentiation. These molecules can be attached to scaffolds or embedded in them to reach the target tissues. Therefore, the functionalization of polymers to load bioactive molecules is also important for enhancing the regeneration of musculoskeletal tissues. However, polymers struggle to accurately control the release of bioactive components. It is also challenging to study the concentrations required for their release to achieve optimal effects.

In this article, we focus on polymer and bioceramic composites for bone tissue repair and polymer-based hydrogel materials for skeletal muscle repair. However, nanosheet materials such as graphene (GN), graphene oxide (GO), and black phosphorus (BP) have gradually become the focus of researchers in recent years because of their unique chemical properties. These materials can be blended with polymers to prepare scaffolds with mechanical strength suitable for musculoskeletal tissue defects, effective induction of host cell proliferation and differentiation, and degradation rate matching the tissue regeneration rate. Additionally, it is difficult to achieve precise control of microarchitecture using conventional manufacturing techniques. By contrast, the 3D bioprinting of customized, personalized scaffolds holds great promise for musculoskeletal tissue regeneration. Compared to other tissue engineering methods, bioprinting permits the control of the pore size, shape, and surface of polymeric materials, as well as the manipulation of various materials, cell types, and bioactive chemicals to replicate the form and function of natural tissues. An understanding of the mechanisms of material-host tissue interactions is crucial for the development of musculoskeletal tissue regeneration and bone engineering.

## Figures and Tables

**Table 1 polymers-14-04566-t001:** Summary of recent studies on polymer-based bioceramic composite scaffolds used in bone tissue engineering.

Scaffold	Structure/Production	Model/Cell	Benefits	Ref.
Collagen/HA	High porous 3D scaffold/ freeze-drying	In vitro study	Enhance mechanical properties; controlled microstructure, porosity, swelling rate, etc.	[111]
Collagen/Mg-HA	Biomimetic 3D scaffold/ freeze-drying	In vitro study/hADSC	Promote bone cell migration, proliferation, and differentiation	[112]
HA/Col/O-CMCS- BMP-2/VEGF	Columnar composite scaffold containing microspheres/extrusion molding, freeze-drying	In vivo study/ rat subcutaneous tissue implantation	Dual factor orderly release of BMP-2 and VEGF promotes collagen organization and neovascularization more effectively	[113]
Collagen-FeMnHA	Lamellar scaffold/ freeze-drying	In vitro and in vivo study/ mouse calvarial defect model	Cellular or lamellar microstructure, Fe/Mn incorporation further amplified the osteogenic promotion	[142]
Chitosan/nHA	Multilayer porous structure/ freeze-drying	In vitro study/MC3T3-E1	Good compressive strength and osteogenic bioactivity	[114]
Chitosan/HA	Hierarchical pore structure/ freeze-drying and cold atmospheric plasma	In vitro study/MC3T3-E1	High porosity (≥80%) and swelling rates (≥300%); excellent cell viability and biomineralization	[115]
Alg-DA/QCHA	Porous gradient scaffold “iterative layering”/ freeze-drying	In vitro and in vivo study/ rabbit femoral defect model	High compression module (1.7 MPa), appropriate degradation rate, and excellent osteogenic activity	[116]
HA/SA/CS@VAN/ VEGF	HA-Reinforced Ball-Bearing Stent/freeze-drying	In vitro study/BMSCs	Chronological release of the two drugs and the good mechanical strength of the scaffold	[117]
nHA/SF	3D porous scaffold/ freeze- drying	In vitro and in vivo study/ rat calvarial critical-size defect model	Good scaffold stability and new bone formation	[118]
HAp/Col/CS	Electrospinning	In vitro study/human fetal osteoblasts	Promote osteoblast proliferation and differentiation as well as mineral deposition	[120]
HAP@CS-PEO@GEL	Coaxial electrospinning technique; a wet chemical method	In vitro study/MG63 cell	Improve the mineralization efficiency of HAP and enhance osteoblast cell proliferation	[143]
PLLA/gelatin/HA	Electrospinning, alternate soaking method	In vivo study/rat calvarial critical-size defect model	High bone mineral density (BMD), and the defect was completely closed at 6 weeks	[122]
Cell-laden collagen/ HA	3D cell-laden porous scaffold/3D bioprinting	In vitro study/MG63 cell and hADSC	Meringue-like porous cell-laden structure; excellent metabolic and osteogenic activities	[144]
PCL/PDA/HA	Stereo lithography	In vitro study/hMSCs	High protein adsorption rate and promotes osteogenic differentiation and angiogenesis	[124]
HA/PCL/CNTs	3D printing	In vitro study/MG63 cell	Good mechanical properties and electrical conductivity; good bioactivity and cell adhesion	[123]
PEEK-HA	Porous cylindrical scaffold/ compounding, granulating and injection molding	In vivo study/cylindrical cavity of the distal femoral epiphysis	The bioactivity was significantly improved, and the tensile strength, fatigue life and Young’s modulus were close to the mechanical properties of cortical bone.	[125]
Collagen/β-TCP	Cerasorb Ortho Foam	In vitro and in vivo study/ rabbit distal femoral condylemodel	Enhance mechanical properties and plasticity, high biocompatibility, and osteoconductivity	[127]
Collagen/β-TCP	Collagen sponge/ freeze-drying, heat dehydration	In vitro and in vivo study/ rat calvarial critical-size defect model	Greater osteoconductivity and better biodegradability than Bio-Oss Collagen	[128]
Chitosan/gelatin/ β-TCP	3D porous scaffold/ freeze- drying	In vitro study/ Human osteoblasts cells (hOB)	Good bioactivity and high extracellular calcium deposition, as well as the bactericidal effect	[130]
Chitosan/gelatin/ β-TCP	3D porous scaffold/ thermally induced phase separation and lyophilization	In vitro study/hMSCs; In vivo study/mouse skin implant model	Promote cell proliferation and MSC differentiation in vitro, and also facilitate the formation of new blood vessels and skin tissue regeneration	[131]
Chitosan/β-TCP@SF	3D porous scaffold/ freeze- drying	In vitro study/MC3T3-E1	Control mechanical properties and hydrophobicity of the scaffold; stimulate the adhesion and proliferation of MC3T3	[145]
SF/β-TCP	3D porous scaffold/ freeze- drying	In vitro and in vivo study/ rat calvarial critical-sizedefect model	Good biocompatibility and highporosity, high compressive strength, and modulus	[132]
PLGA/TCP/Mg	Low-temperature rapid prototyping technology	In vitro and in vivo study/ steroid-associated osteonecrosis rabbit model	Good biomimetic structural design and mechanical properties; excellent osteogenic and angiogenic properties	[133]
Collagen-nBG	Cylindrical gel rolls/ plastic compression technique, freeze-drying	In vitro study/MC3T3-E1	Good mineralizability and compression modulus and promote MC3T3-E1 metabolic activity	[136]
Co^2+^-BG/collagen- glycosaminoglycan	Highly porous scaffold/ freeze-drying	In vitro study/HUVEC	High compressive modulus and porosity (>97%); support osteoblast proliferation and differentiation; excellent angiogenic capacity	[137]
58S BG@Alg	3D porous scaffold/ foam replication method	In vitro study/hMSCs	Improve mechanical properties, antimicrobial activity, and osteogenic differentiation	[138]
45S5BG@Mn-MBGNs/gelatin	3D porous scaffold/ foam replication method, dip coating	In vitro study/MG63 cell	Increase compressive strength and high porosity (>80%); promote cell proliferation	[139]
Collagen-MBGNF	Electrospinning/ freeze-drying	In vitro study/MG63 cell	Excellent osteogenic and mineralization abilities	[141]
BG-HA-PLLA	Electrospinning	In vivo study/rat calvarial defect model	Almost complete regeneration of the calvarial defect after 8 weeks	[140]

**Table 2 polymers-14-04566-t002:** Summary of recent studies on polymer-based hydrogel materials in skeletal muscle tissue engineering.

Scaffold	Structure/Production	Model/Cell	Benefits	Ref.
Collagen	Hydrogel	In vitro and in vivo study/single rat muscle defect model of the entire biceps femoris/rat skeletal myoblasts	Exhibit a large number of branching microvascular networks in vitro	[158]
Gelatin	μmolded gelatin hydrogel cross-linked with MTG	In vitro study/C2C12 cell	Show higher myogenic index, myotube width, and myotube length	[159]
Gelatin	3D printing, in situ printing	In vitro and in vivo study/Quadriceps muscle defect in C57/Bl6 mice/C2C12 cell	Formation of multinucleated myotubes after 24 days, which supports myogenesis and promotes muscle hypertrophy after injury	[160]
GelMA-alginate	Hydrogel/3D printing	In vitro study/C2C12 cell	Enhance the viability and metabolic activity of C2C12 cells	[161]
Hyaluronic acid	Hydrogel	In vitro study/skeletal muscle myoblasts	Have good compatibility with skeletal muscle myogenic cells, which facilitate cell adhesion	[162]
Hyaluronic acid	Photopolymerized hydrogel	In vivo study/mouse tibialis anterior muscle defects	Significantly improve the number of newborn muscle fibers and muscle structures	[163]
Hyaluronic acid /laminin-111 /muscle graft	Hydrogel	In vivo study/VML defects in rat tibialis anterior muscle	The regeneration of new muscle fibers increased, and some muscle function was also restored	[164]
RGD-alginate	Hydrogel	In vitro and in vivo study/subcutaneous implantation model/ gingival mesenchymal stem cell (GMSC)	Higher expression of GMSC myogenic differentiation mRNA and greater myogenic regenerative capacity compared to hBMSC	[165]
Alginate-borax	Hydrogel	In vitro and in vivo study/acute injury model of cardiotoxin in mice/C2C12 cell	Enhance myotube formation, improve and accelerate muscle repair	[167]
Alginate-gelatin/ skeletal muscle ECM	Hydrogel	In vitro study/human skeletal muscle progenitor cells (hSMPCs)	Significantly enhance cell expansion, differentiation, and maturation of hSMPC	[176]
Chitosan	Hydrogel	In vivo study/rat skeletal muscle injury model	Enable C2C12 cells delivery and promote skeletal muscle regeneration	[168]
Chitosan-gelatin	Hydrogel/3D printing	In vivo study/L929 cell	Chitosan-gelatin hybrid hydrogel bioink with good cytocompatibility and stability for tissue regeneration	[169]
PEG-4MAL/MuSC	Hydrogel	In vivo study/mouse model of hypothermic injury to the anterior tibial muscles/MuSC	Significantly improve in vivo cell survival, proliferation, and implantation	[170]
PEG-4MAL/MuSC /Wnt7a	Hydrogel	In vivo study/mouse model of hypothermic injury to the anterior tibial muscles/MuSC	Significantly promote MuSC migration and muscle fiber hypertrophy	[171]
PLA-aniline tetramer	Hydrogel	In vitro study/C2C12 cell	Obtain electrically conductive, ductile synthetic hydrogels; significantly promote the proliferation and myogenic differentiation of C2C12 cells	[172]
PEG-LM111/ PEGDA	Hydrogel	In vitro study/C2C12 cell	Promote myoblast adhesion, survival, and growth factor production	[90]
PCL/SF/PANI- PEGS-M	Core-shell composite scaffold-hydrogel shell	In vitro study/C2C12 cell	Effectively induce myogenic differentiation and myotube formation of C2C12 myogenic cells	[174]
Methacrylic acid- collagen	Hydrogel	In vivo study/mouse tibialis anterior muscle defects	Significantly improve the quality and function of the regenerated muscle muscles; improve vascularization and innervation	[175]

## Data Availability

Not applicable.

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
