# Peer review of "Recent Advances in the Application of Natural and Synthetic Polymer-Based Scaffolds in Musculoskeletal Regeneration"

_polymers, 2022, doi:10.3390/polym14214566_

Round 1
Reviewer 1 Report
Manuscript title: Recent Advances in the Application of Natural and Synthetic
Polymer-based Scaffolds in Musculoskeletal Regeneration
Manuscript ID: polymers 1991519
Comments:
This is an interesting and considerable comprehensive review. B. Ye et al have attempted to review on the natural and synthetic polymer based scaffolds for musculoskeletal regeneration and related applications. The authors have conducted quite an extensive review pertaining to this topic, specifically on various types of natural and synthetic polymer. The authors also include the discussion of specific fabrication methods, i.e. electrospinning and additive manufacturing for polymer-based composite scaffolds. However, I find that some points needed to be addressed to improve the manuscript further. This work is suitable to be published in polymers journal. In my opinion, this work can be accepted for publication after the following minor corrections are made.
1. Novelty and significance of manuscript: I suggest the authors should further highlight the novelty and significance of manuscript to demonstrate the worthiness of publishing this manuscript in the introduction part. The current version: “In this review, we summarize the different types of natural and synthetic polymeric 81 biomaterials used for tissue engineering applications and review the recent advances in 82 polymer-based scaffold materials for musculoskeletal tissue engineering, mainly includ-83 ing polymer-based bioceramic composite scaffolds for bone tissue repair and polymer-84 based hydrogels for skeletal muscle tissue repair” seems to portray merely a summary without any critical review. The authors should re-write this last paragraph in the introduction section.
2. Section 4.1.3.4: Are electrospinning and 3D additive manufacturing the only two methods use for the fabrication of polymer-based composite scaffolds? Please consider to expand the write-up on the methods of fabrication aspect.
3. English problem: I suggest the authors to check the English language by native English speakers.
Author Response
Manuscript Number: polymers-1991519
Response to the comments of the Reviewer
Dear editor and reviewers:
Thank you for your letter and for the reviewer’s comments concerning our manuscript entitled “Recent Advances in the Application of Natural and Synthetic Polymer-based Scaffolds in Musculoskeletal Regeneration”, manuscript Number: polymers-1991519. We highly appreciate these comments! Overall, the comments have been fair, encouraging and constructive. They are very helpful for revising and improving our paper. We have studied comments carefully and have made corrections which we hope meet with approval. Revised portions are highlighted in the text hightlight color on the paper. English revisions are viewed through "Track Changes". The main corrections in the paper and the response to the reviewer’s comments are as following:
Reviewer #1: This is an interesting and considerable comprehensive review. B. Ye et al have attempted to review on the natural and synthetic polymer-based scaffolds for musculoskeletal regeneration and related applications. The authors have conducted quite an extensive review pertaining to this topic, specifically on various types of natural and synthetic polymer. The authors also include the discussion of specific fabrication methods, i.e. electrospinning and additive manufacturing for polymer-based composite scaffolds. However, I find that some points needed to be addressed to improve the manuscript further. This work is suitable to be published in polymers journal. In my opinion, this work can be accepted for publication after the following minor corrections are made.
Response: We appreciate your positive review and constructive comments.
Comment 1: Novelty and significance of manuscript: I suggest the authors should further highlight the novelty and significance of manuscript to demonstrate the worthiness of publishing this manuscript in the introduction part. The current version: “In this review, we summarize the different types of natural and synthetic polymeric 81 biomaterials used for tissue engineering applications and review the recent advances in polymer-based scaffold materials for musculoskeletal tissue engineering, mainly including polymer-based bioceramic composite scaffolds for bone tissue repair and polymer-based hydrogels for skeletal muscle tissue repair” seems to portray merely a summary without any critical review. The authors should re-write this last paragraph in the introduction section.
Response: Thank for the comment. We wrote the last paragraph of the introduction section too concisely and did not highlight the novelty and significance of the manuscript, so we rewrote the last paragraph in conjunction with the revised section, detailing the structure of our entire review, the development of the narrative, and highlighting his significance. This revised section is as follows:
“Polymeric materials have a wide range of promising applications in orthopedics, and various types of polymer-based scaffolds have been developed for musculoskeletal tissue repair, but the relevant reviews are not presented comprehensively enough. Therefore, this review first look at different types of natural and synthetic polymeric biomaterials for tissue engineering applications, then describes the structural and re-generative properties of bone and skeletal muscle, presents the scaffold requirements for musculoskeletal tissue engineering, and provides a systematic review of polymer-based scaffolds suitable for bone and skeletal muscle regeneration. Finally, this review briefly discusses various fabrication techniques used for polymer processing as well as current challenges and future prospects for musculoskeletal tissue engineering. We expect this review to provide a reference for colleagues engaged in research related to polymeric materials for musculoskeletal regeneration and to promote the progress and development of musculoskeletal tissue engineering.
Comment 2: Section 4.1.3.4: Are electrospinning and 3D additive manufacturing the only two methods use for the fabrication of polymer-based composite scaffolds? Please consider to expand the write-up on the methods of fabrication aspect.
Response: Thank for the suggestive comment. There are several fabrication methods for polymer composite scaffolds, and our original manuscript was not comprehensive enough, mainly introducing the emerging technologies of electrospinning and 3D additive manufacturing. Now, we have replaced Section 4.1.3.4 with Section 5, which discusses common polymer scaffold fabrication methods, expanding on such methods as Solvent Casting and Particulate Leaching (SCPL), Gas Foaming, Thermally Induced Phase Separation (TIPS), and Freeze-Drying.
Comment 3: English problem: I suggest the authors to check the English language by native English speakers.
Response: Thank you for your suggestion. We carefully reviewed the English grammar and vocabulary in the article and made changes during the revision process. In addition, we contacted a professional organization to make revisions to this review to ensure the correctness and fluency of the article. We upload the revision certificate in the attachment. English revisions are viewed through "Track Changes".

Reviewer 2 Report
The manuscript contains a review of the results of scientific research in recent years, published in 171 articles, concerning the application of natural and synthetic polymeric composites and combinations of polymeric and inorganic materials for musculoskeletal tissue repair. The authors summarize the results of the analysis of achievements in a number of topical areas of natural and synthetic polymeric composites in musculoskeletal tissue engineering and musculoskeletal rehabilitation and treatment.
The results of the analysis of the results in the reviewed field of scientific researches allowed specifying current problems and future research perspectives.
The results of the summary are of interest to a wide range of specialists who carry out research in the topical scientific area related to the creation of natural and synthetic polymeric composite materials for bone and muscle tissue repair, as well as graduate and postgraduate students of universities.
Unfortunately, the review did not include results of publications related to achieving the most important characteristics of biocompatible materials, which include strength and functional properties (elasticity moduli, strength characteristics, tribological characteristics, wear resistance, fatigue life, etc.). The requirements for musculoskeletal engineering materials include not only biocompatibility issues, but also requirements of durability and mechanical compatibility natural and synthetic polymeric composites with musculoskeletal tissue.

Author Response
Manuscript Number: polymers-1991519
Response to the comments of the Reviewer
Dear editor and reviewers:
Thank you for your letter and for the reviewer’s comments concerning our manuscript entitled “Recent Advances in the Application of Natural and Synthetic Polymer-based Scaffolds in Musculoskeletal Regeneration”, manuscript Number: polymers-1991519. We highly appreciate these comments! Overall, the comments have been fair, encouraging and constructive. They are very helpful for revising and improving our paper. We have studied comments carefully and have made corrections which we hope meet with approval. Revised portions are highlighted in the text hightlight color on the paper. English revisions are viewed through "Track Changes". The main corrections in the paper and the response to the reviewer’s comments are as following:
Reviewer #2:
The manuscript contains a review of the results of scientific research in recent years, published in 171 articles, concerning the application of natural and synthetic polymeric composites and combinations of polymeric and inorganic materials for musculoskeletal tissue repair. The authors summarize the results of the analysis of achievements in a number of topical areas of natural and synthetic polymeric composites in musculoskeletal tissue engineering and musculoskeletal rehabilitation and treatment.
The results of the analysis of the results in the reviewed field of scientific researches allowed specifying current problems and future research perspectives.
The results of the summary are of interest to a wide range of specialists who carry out research in the topical scientific area related to the creation of natural and synthetic polymeric composite materials for bone and muscle tissue repair, as well as graduate and postgraduate students of universities.
Response: We appreciate your positive review and constructive comments.
Comment: Unfortunately, the review did not include results of publications related to achieving the most important characteristics of biocompatible materials, which include strength and functional properties (elasticity moduli, strength characteristics, tribological characteristics, wear resistance, fatigue life, etc.). The requirements for musculoskeletal engineering materials include not only biocompatibility issues, but also requirements of durability and mechanical compatibility natural and synthetic polymeric composites with musculoskeletal tissue.
Response: Thank for the suggestive comment. We develop polymer composites for musculoskeletal tissue engineering. The elastic modulus and mechanical strength of the composites are particularly important as weightbearing and moving parts. We are also focusing on this in our review of related research, which is presented in the following sections of the article.
(a) section 4.1.3.1 line 292-293: “The addition of HA significantly increased the compression modulus of collagen scaffolds”
(b) section 4.1.3.1 line 305-306: “Compared with the pure chitosan scaffold, the composite had good compressive strength”
(c) section 4.1.3.1 line 313-314: “The scaffolds exhibited a high compression modulus (1.7 MPa) ”
(d) section 4.1.3.1 line 324: “silk-fabricated HA scaffolds with high compressive strength and porosity”
(e) section 4.1.3.2 line 373-374: “By incorporating 10 wt% to 30 wt% of β-TCP content, the compressive strength of the composites ranged from 0.8 MPa to 2.45 MPa”
(f) section 4.1.3.3 line 397-398: “the compressive modulus of the scaffold increased thirteen-fold by day7”
(g) section 4.1.3.3 line 400-401: “Quinlan et al. fabricated a Co2+-BG/collagen-glycosaminoglycan scaffold with a high compressive modulus and high porosity (>97%)”
But at the same time we do focus mainly on biocompatibility and bioactivity and not enough on strength and functional properties. Different material application scenarios have different material requirements, and the frictional properties, wear resistance, and fatigue life of biomaterials are also important.
Just as PEEK has a wide range of applications in orthopedic implants, such as the already used intervertebral fusion device CAGE, PEEK also has good physical and chemical stability, such as wear resistance, fatigue resistance, etc., and is promising in bone tissue reconstruction. Therefore, we have added section 3.6 Polyetheretherketone (PEEK) to section 3 Synthetic Polymers. We present these properties of material as a representative of PEEK and discuss its application in composite materials.
In the future, we will pay more attention to the strength and functional properties of biomaterials in the development of musculoskeletal engineering materials, and we thank you again for your suggestions.
